# A rapid, accurate, scalable, and portable testing system for COVID-19 diagnosis

Guanhua Xun [1,5], Stephan Thomas Lane[2,5], Vassily Andrew Petrov[2], Brandon Elliott Pepa[3] & Huimin Zhao [1,2,4✉]

The need for rapid, accurate, and scalable testing systems for COVID-19 diagnosis is clear and urgent. Here, we report a rapid Scalable and Portable Testing (SPOT) system consisting of a rapid, highly sensitive, and accurate assay and a battery-powered portable device for COVID-19 diagnosis. The SPOT assay comprises a one-pot reverse transcriptase-loop-mediated isothermal amplification (RT-LAMP) followed by *Pf*Ago-based target sequence detection. It is capable of detecting the N gene and E gene in a multiplexed reaction with the limit of detection (LoD) of 0.44 copies/μL and 1.09 copies/μL, respectively, in SARS-CoV-2 virus-spiked saliva samples within 30 min. Moreover, the SPOT system is used to analyze 104 clinical saliva samples and identified 28/30 (93.3% sensitivity) SARS-CoV-2 positive samples (100% sensitivity if LoD is considered) and 73/74 (98.6% specificity) SARS-CoV-2 negative samples. This combination of speed, accuracy, sensitivity, and portability will enable high-volume, low-cost access to areas in need of urgent COVID-19 testing capabilities.

---

[1] Department of Bioengineering, University of Illinois at Urbana-Champaign, Urbana, IL, USA. [2] Carl R. Woese Institute for Genomic Biology, University of Illinois at Urbana-Champaign, Urbana, IL, USA. [3] Department of Mechanical Engineering, University of Illinois at Urbana-Champaign, Urbana, IL, USA. [4] Departments of Chemical and Biomolecular Engineering, Chemistry, and Biochemistry, University of Illinois at Urbana-Champaign, Urbana, IL, USA. [5]These authors contributed equally: Guanhua Xun, Stephan Thomas Lane. ✉email: zhao5@illinois.edu

Rapid, accurate, and scalable testing systems for COVID-19 diagnosis are essential to control SARS-CoV-2, the novel coronavirus causing COVID-19 pandemic, and reopen society. The diagnosis of COVID-19 mainly relies on the detection of the coronavirus RNA. The gold standard testing system approved by US Centers for Disease Control and Prevention (CDC) comprises a thermocycler and a bioassay based on the quantitative reverse transcriptase-polymerase chain reaction (qRT-PCR)[1]. Other existing testing systems are based on reverse transcriptase-loop-mediated isothermal amplification (RT-LAMP)[2]. For example, colorimetric LAMP takes advantage of the drop in pH during the LAMP reaction to induce a change in solution color from pink to yellow after successful nucleic acid amplification[3,4]. DNA Endonuclease-Targeted CRISPR Trans Reporter couples LbCas12a target sequence activated ssDNase activity with RT-LAMP amplification to detect specific sequences by fluorescence or lateral flow[5]. Similarly, Specific High Sensitivity Enzymatic Reporter UnLOCKing harnesses the collateral DNase activity of AapCas12b[6] together with RT-LAMP to produce a visible readout on a lateral flow strip after collateral cleavage of reporters upon activation by SARS-CoV-2 nucleic acid amplicons. Although these RT-LAMP-based testing systems have eliminated the requirement for expensive and bulky thermocyclers, they have only limited portability due to their requirements for heat blocks or water baths.

Both qRT-PCR and RT-LAMP-based assays have their advantages and disadvantages. The former is widely used for virus identification with high sensitivity and specificity, but its analysis requires various equipment and educated analysts, which is only possible in a laboratory[7]. It normally takes 4–6 h for testing and over 24 h to return the result to the patient[5]. The latter is a highly sensitive nucleic acid amplification method that can often detect small numbers of DNA or RNA templates within half an hour[8,9]. However, the requirement for high temperature limits its compatibility with other enzymes, forcing CRISPR-Cas-based methods to run amplification and detection steps separately or optimize the Cas protein to survive at high temperatures[6]. Although some Cas proteins can function at 60 °C, this temperature is suboptimal for both the Cas protein and Bst 2.0, the DNA polymerase used in LAMP, dramatically decreasing their activity and prolonging the detection period. Separating the amplification and detection steps into a two-step reaction increases the risk of contamination due to the extremely high sensitivity of RT-LAMP. Moreover, existing RT-LAMP-based methods can only detect one gene at a time, reducing the accuracy of diagnosis. Most importantly, because all current COVID-19 tests require many complicated operation steps, scaling up requires trained personnel and specialized tools.

Here we report a rapid Scalable and Portable Testing (SPOT) system consisting of a rapid, highly sensitive, and accurate assay and a battery-powered portable device for COVID-19 diagnosis. This device consists of 3D printed casing and internal structure with precise temperature control and fluorescence detection, whereas this assay combines RT-LAMP with an Argonaute protein from hyperthermophilic archaeon *Pyrococcus furiosus* (*Pf*Ago) capable of precise recognition and cleavage of a target DNA at 95 °C as directed by small 5′-phosphorylated single strand DNA (ssDNA) as guide DNA (gDNA)[10,11]. Due to the multi-turnover activity of *Pf*Ago, its secondary cleavage mechanism can be harnessed for specific, sensitive, and multiplex nucleic acid detection[12,13]. For COVID-19 samples, although nasopharyngeal swab and nasal swab samples were recommended for detection of SARS-CoV-2, saliva samples are a more attractive alternative due to the ease, safety, and non-invasive nature of its collection[14,15], and its relatively high viral load during the first week of infection[16]. These benefits enable a saliva sample to be an ideal specimen for reliable and rapid self-detection without professional supervision[17–19]. While current CRISPR-based detection systems normally require 50 min for testing, *Pf*Ago can dramatically speed up the detection process by requiring only 3–5 min for cleavage of amplified products, thereby shortening the total turnaround time for testing to less than 30 min. Moreover, successful *Pf*Ago detection requires at least two sequence-specific cleavages, endowing the SPOT system with high specificity and the ability for multiplexing. Finally, to validate the SPOT system, the sensitivity and accuracy of the SPOT system were determined by using 104 clinical saliva samples.

## Results

**Design and optimization of the SPOT assay**. The entire workflow of the SPOT system is shown in Fig. 1a. Briefly, pretreated saliva samples will be tested automatically in a portable device which includes precise heating and optical modules, producing fluorescent output signals upon detection of SARS-CoV-2 nucleic acid sequences. Initially, the saliva sample collected by a capillary will be mixed with QuickExtract DNA extraction solution in a sample preparation capillary (capillary 1) for 5 min heat pretreatment. Using a transfer capillary, users inject pretreated saliva samples into the assay capillary (capillary 2) containing reagents for viral RNA amplification and detection. Detection will then be initiated by switching on the device: the heating module will first hold at 63 °C for 20–30 min for LAMP amplification followed by automatically heating to 95 °C for 3–5 min for *Pf*Ago detection, and finally the LED excitation module will shine through the capillary. Fluorescence is detected by photodiodes secured behind bandpass filters specific for emissions of the chosen fluorophores and test results are displayed on an attached LCD screen. The SPOT device is calibrated using negative control samples to quantify the background signal with the positive threshold set to 5 × s.d. above background in the device's program.

To develop the assay for COVID-19 diagnostics using the SPOT system, we used RT-LAMP to amplify the E (envelope) and N (nucleoprotein) genes of SARS-CoV-2. A set of gDNAs induces targeted *Pf*Ago cleavage on the amplicon to release a 16 nucleotide (nt) ssDNA fragment, which acts as a secondary gDNA, targeting *Pf*Ago cleavage to the specific Fluorophore-Quencher (F-Q) labeled ssDNA reporter probe (Fig. 1b, Fig. S1). Since the reagents may affect the structure of the reporter and overall fluorescent background signal, we optimized the GC content and length of the reporter to minimize the unwanted background fluorescence (Fig. S2a). We found that 17 nt reporters, the minimum size necessary to enable secondary cleavage, with low G-C content and labeled with a 5′-fluorophore and 3′-quencher produced relatively low background signals. We compared the cleavage efficiency in target gene detection reactions containing two or three primary gDNAs and found use of three gDNAs enhanced the efficiency of cleavage, which led to a 2–3 fold higher fluorescent signal than use of only two gDNAs (Fig. S2b).

**Analysis of LoD and specificity**. We next determined the limit of detection (LoD) of the SPOT assay by spiking saliva samples with synthetic in vitro-transcribed (IVT) SARS-CoV-2 RNA followed by RT-LAMP and *Pf*Ago-based detection reactions. After 20 min LAMP amplification at 63 °C, the *Pf*Ago detection components were added into the reaction mixture followed by heating at 95 °C for 3 min and quantification of readouts with a fluorometer. We found that the LoD of the SPOT assay is ~7.5 copies/reaction (0.19 copies/µL) with in vitro-transcribed RNA samples (Fig. 2a). To determine the feasibility of the assay for detection of SARS-CoV-2 in real saliva samples, we next assayed saliva samples spiked with

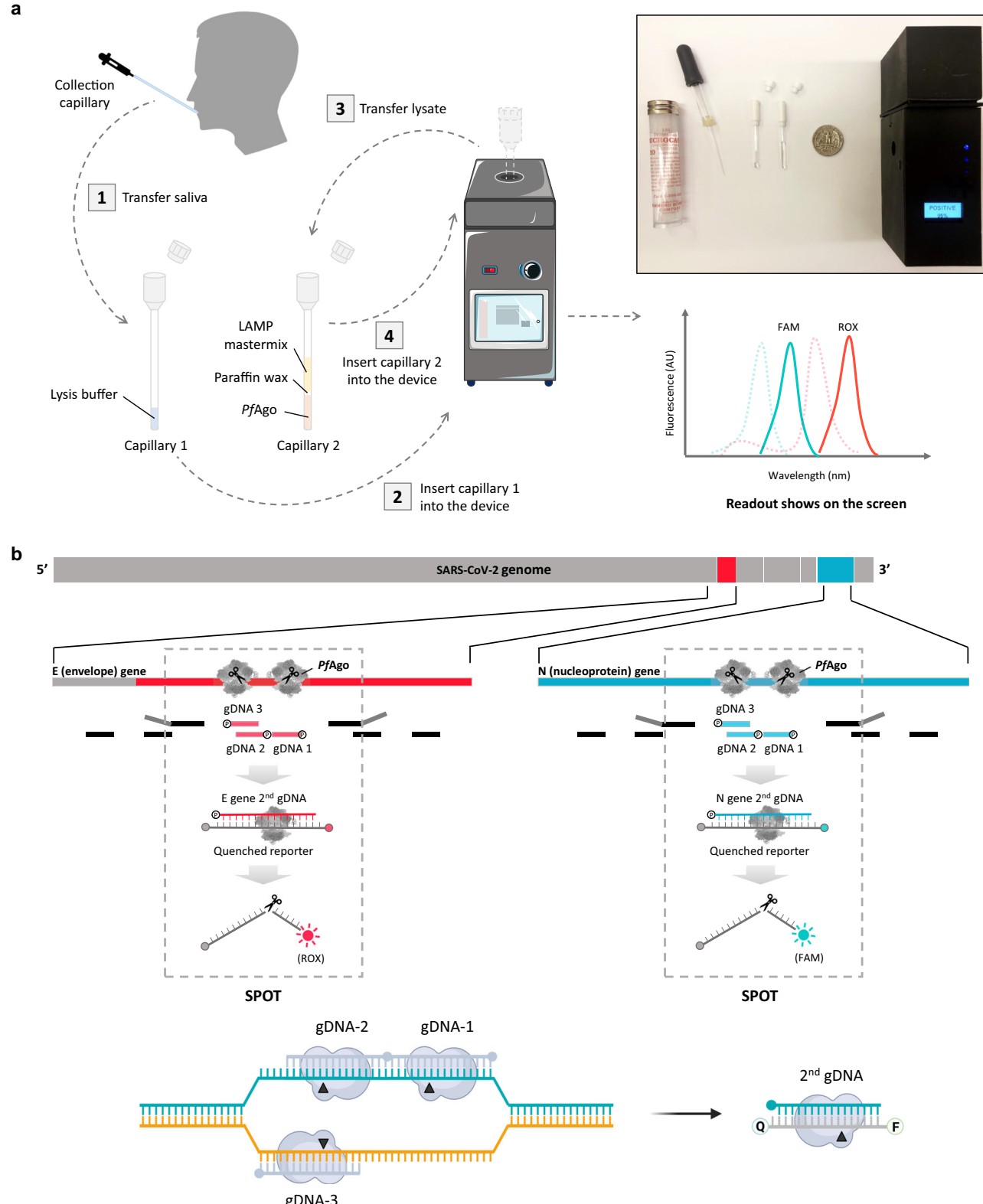

**Fig. 1 Overview of the SPOT system. a** Overall workflow using capillaries and the SPOT device. Green and red dashed lines represent excitation spectrum of FAM and ROX, respectively, while emission spectrums are shown with green and red solid lines. *Inset*: Device and consumables needed to run the SPOT system, including a 3D printed portable detection device, a prefabricated sample pretreatment capillary 1, a prefabricated viral RNA detection capillary 2, and collection capillaries. A quarter coin is placed for scale (diameter: 24.26 mm); **b** Genome map showing primers, gDNAs, and SPOT mechanism. RT-LAMP primers are indicated by black rectangles. Enclosed "P" represents the phosphate group. *Pf*Ago cleavage sites are shown in black triangles, enclosed "Q" represents quencher group and enclosed "F" represents fluorophore group.

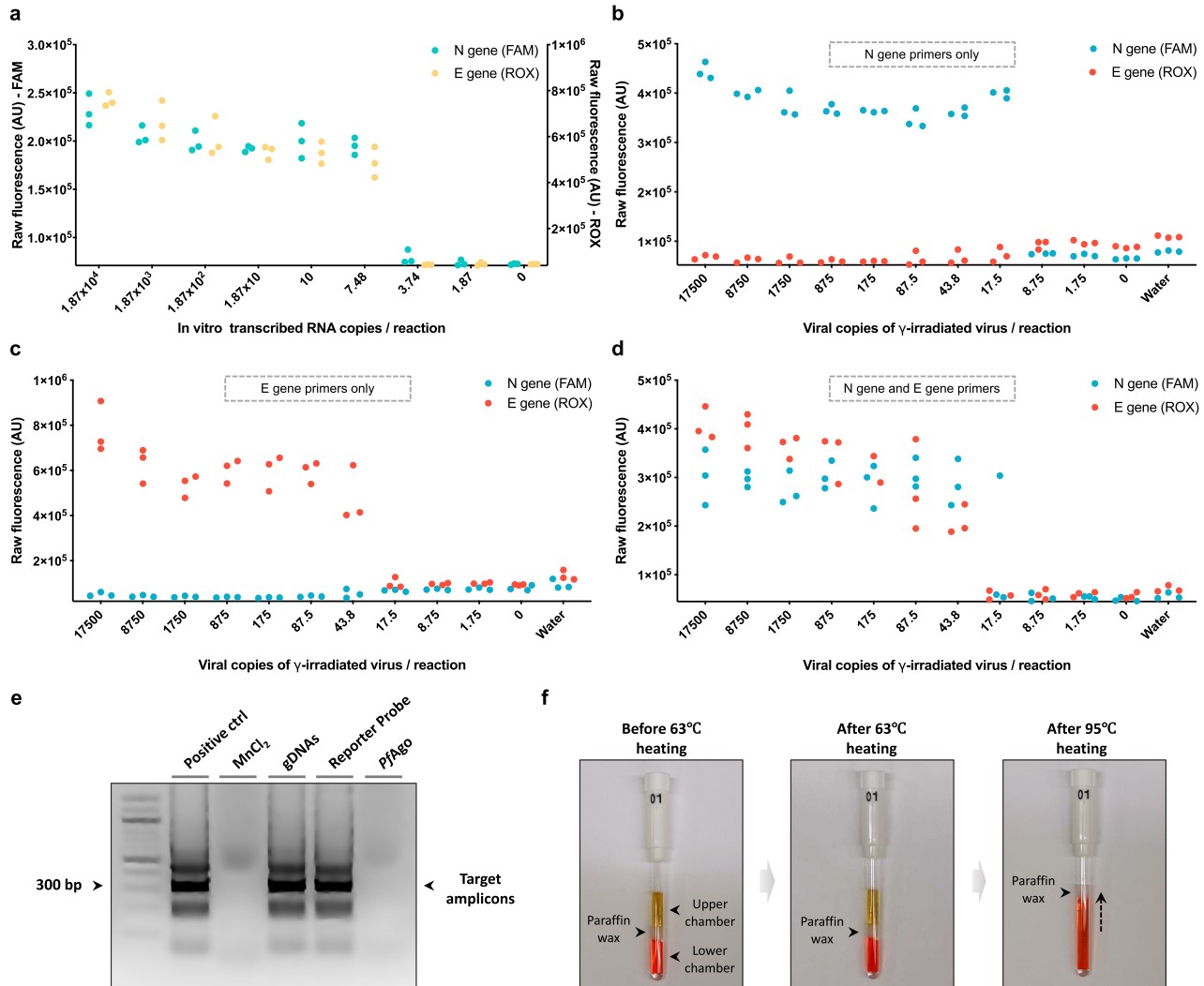

**Fig. 2 Optimization and validation of the SPOT assay. a** LoD using RNA samples. Reactions were performed in a thermocycler and fluorescence values were measured on a qPCR machine ($n = 3$) using SARS-CoV-2 N and E genes IVT RNAs; **b** LoD using virus-spiked saliva samples with only N gene corresponding primers. Reactions were performed in a thermocycler and fluorescence values were measured using a qPCR machine ($n = 3$); **c** LoD using virus-spiked saliva samples with only E gene corresponding primers. Reactions were performed in a thermocycler and fluorescence values were measured using a qPCR machine ($n = 3$); **d** Multiplexity of the assay. Both N gene and E gene corresponding primers were added. Reactions were performed in a thermocycler and fluorescence values were measured using a qPCR machine ($n = 3$); **e** *Pf*Ago detection components affect LAMP reaction. Lanes from left to right: LAMP positive control, 0.5 mM manganese ion, 1 mM gDNAs, 0.3 μM fluorophore-quencher labeled reporter probe, 5.5 μM *Pf*Ago added (this experiment was repeated three times with similar results); **f** Strategy for compartmenting two reactions: the upper chamber (yellow liquid represents RT-LAMP reaction) and the lower chamber (red liquid represents mixture of *Pf*Ago and manganese ion) are separated by paraffin wax (melting point above 70 °C). Source data are provided as a Source Data file.

γ-irradiated SARS-CoV-2. The virus-spiked saliva samples were first pre-treated by adding QuickExtract DNA Extraction Solution and heating at 95 °C for 5 min (Figs. S5 and S6). The LoD was as low as 17.5 copies per reaction (0.44 copies/μL). Extending the LAMP reaction to 30 min is recommended to detect SARS-CoV-2 in samples with an original viral number below 50 copies per reaction (Fig. S3). Based on the time-course assay, we found that 5 min *Pf*Ago cleavage is sufficient to obtain the obvious fluorescent signal increases for detecting contrived saliva samples with the LoD viral load (Fig. S4).

To achieve high accuracy of our assay, we next sought to detect two viral genes simultaneously in the same reaction targeting N gene region (US CDC assay) and E gene region (World Health Organization assay)[1,20]. We designed two sets of gDNAs targeting the N and E genes with two different fluorophores to establish a one-pot multiplexed reaction. When the N or E gene related primers were added into reaction separately, we only observed the specific fluorescence accordingly, showing that *Pf*Ago mediated multiplexed reaction is carried out in an orthogonal manner. Our method can detect the N and E genes in a multiplexed reaction with a LoD of 17.5 copies (0.44 copies/μL, Fig. 2b) and 43.8 copies (1.09 copies/μL, Fig. 2c), respectively, using virus-spiked saliva samples. We then added two sets of primers in one reaction, and the result showed that both viral genes can be amplified and detected successfully (Fig. 2d). In order to validate the specificity of our detection system to SARS-CoV-2, saliva was spiked with or without SARS-CoV-2, three other human coronaviruses (OC43, 229E, and NL63) genomic RNA, SARS and MERS viruses (γ-irradiated), and influenza A genomic RNA. Among these samples, SARS-CoV-2 genes were only detected in the positive control, further supporting the specificity of the detection platform for SARS-CoV-2 (Fig. S8).

**Development of the one-pot SPOT reaction cascade**. To perform one-pot amplification and detection, some optimizations are required as components of the LAMP and *Pf*Ago detection reactions are incompatible. The 5′-3′ dual-phosphorylated primary gDNAs used to mediate primary cleavage prevent priming during LAMP. Additionally, manganese ions ($Mn^{2+}$), which are key to *Pf*Ago cleavage activity, inhibit the LAMP reaction and *Pf*Ago itself can partially inhibit amplification by tightly binding to DNA (Fig. 2e). Thus, we developed an encapsulation method to isolate the LAMP and *Pf*Ago reaction components. The *Pf*Ago reaction mixture was loaded into the capillary (diameter = 3.18 mm), sealed with paraffin wax, followed by loading the LAMP reaction mixture to the upper chamber of the capillary. After heating above 70 °C, the paraffin wax melts, releasing the *Pf*Ago reaction mixture and initiating the detection process (Fig. 2f).

**Design and construction of the SPOT device**. Next, we designed and built a prototype portable device capable of performing sample pretreatment, the assay, and quantifying the fluorescent output (Fig. 3a). This prototype device consists of 3D printed casing and internal structure with temperature control accomplished via a copper heat block wrapped with nichrome wire. An integrated fan enables rapid cooling of the sample capillary after completion of pretreatment or detection reactions. Fluorescent excitation is accomplished using a single blue LED aimed into the bottom of the capillary with fluorescent emissions detected by photodiodes placed behind fluorophore-specific emission filters. Temperature is controlled and fluorescence is quantified by three custom printed circuit boards: an LED board, a photodiode board, and a motherboard equipped with a microcontroller. The front of the device contains buttons for powering the device, starting or resetting the assay, and an LCD screen to display results. The system is powered by a rechargeable and swappable lithium-ion battery. This assembled prototype was named as SPOT device and directly compared with commercial machines in its ability to perform temperature control and fluorescence quantification.

To evaluate the accuracy and stability of its heating module, reactions were conducted using both the SPOT device and a thermocycler. We compared the fluorescence readouts obtained from the SPOT device and a qPCR machine (QuantStudio 3 Real-Time PCR system), demonstrating that the SPOT device is as capable as a commercial thermocycler at performing the SPOT assay (Fig. 3b, c, Fig. S9). Moreover, a comparison of end-point fluorescence readouts from the SPOT device and a qPCR machine shows that the SPOT device is sensitive enough to accurately replicate the fluorescent measurements obtained from the qPCR machine (Fig. S10).

**Detection of SARS-CoV-2 from clinical saliva samples using the SPOT system**. We evaluated the performance of the SPOT system in a blinded test using 104 clinical saliva samples obtained from the Veterinary Diagnostic Laboratory at the University of Illinois at Urbana-Champaign. These samples were analyzed independently by scientists not involved in this study using the qRT-PCR approach and three target genes: ORF1ab gene (encoding a replication protein), S gene (encoding the spike protein), and N gene (encoding the nucleoprotein). By comparison, we were able to reliably detect 28 out of 30 (93.3%) SARS-CoV-2 positive patient samples (both N and E genes were detected) and 72 out of 74 (97.3%) SARS-CoV-2 negative samples (neither N nor E genes were detected). Two samples were declared inconclusive as only one of the two target genes was detected. After re-testing, one of these inconclusive samples was found to be negative but the other remained inconclusive

(Fig. 3d). The two missed positive samples each had a Ct value of above 31 cycles for N gene and 32.3–36.6 cycles for S and ORF1ab genes. These Ct values correspond to original sample viral copy numbers of $5 \times 10^2$–$2.5 \times 10^3$ copies/mL[16], which are below the LoD of the SPOT system when using saliva samples. However, when using in vitro transcribed RNA samples, the LoD of the SPOT system matches the LoD of the CDC's gold standard qRT-PCR testing system[21], indicating that optimization of the sample pretreatment method may increase the sensitivity of the SPOT assay.

## Discussion

One key innovation of the SPOT system is the high specificity and accuracy resulting from *Pf*Ago's unique multiplexing capability. LAMP reactions are prone to non-specific amplification, significantly increasing the rate of false positives and precluding their use as a diagnostic assay. Coupling LAMP with Cas12-based nucleic acid detection reduces these false positives by requiring a single sequence-specific cleavage to generate an output signal[5]. The *Pf*Ago-based detection method further improves upon specificity by requiring multiple sequence-specific cleavages to generate a positive test result: two cuts with primary guides to release the 16 nt secondary guide, followed by secondary cleavage of the quenched probe. Moreover, Cas12-based methods rely upon collateral activity to indiscriminately cleave quenched probes after recognition of a target site and cannot be used to link multiple specific recognition sites to distinct output signals in a single reaction[5,6]. In contrast, the sequence-specific cleavage of a quenched probe by *Pf*Ago in the SPOT assay enables multiplexing by linking a unique secondary gDNA to the release of a fluorophore. Multiplexing can not only increase the specificity of detection, but also enable testing for multiple viruses and bacteria or distinguishing between SARS-CoV-2 variants in a single reaction.

A second key innovation of the SPOT system is the development of a portable, hand-held, and battery-powered testing device. The SPOT device is assembled from simple, readily available components: electronics, optical filters, and a heat block. The material costs would sum to less than $78 per device at scale production of 10,000 devices (Supplementary Data 2). The reagents and consumables for each test total to only $6.14, clearing the way for low-cost portable SARS-CoV-2 diagnostics. Testing results from the SPOT device can be transferred to a computer or smartphone with a USB cable, enabling centralized data management to support test-trace-isolate protocols. Considering that high testing frequency coupled with quick turn-around time may be more important than test sensitivity to efficiently combat spread of the SARS-CoV-2 virus[22], portable and low-cost testing systems such as the SPOT system will play a critical role in deploying large-scale rapid and portable diagnostics for this and future pandemics.

The SPOT system presented here enables rapid diagnosis without trained personnel and only minimal liquid handling. While the current prototype has a capacity of only one reaction at a time, an enlarged heating system and separated module for fluorescent quantification is a clear path to building a low-cost high throughput SPOT device (~1000 tests per day per testing system). The SPOT system has great potential to enable rapid and low-cost test-trace-isolate measures with minimal financial investment or training requirements, expanding the availability of testing and serving as a key tool to fight the COVID-19 pandemic. By connecting the SPOT system to the internet, an advanced real-time data upload system could endow the SPOT system with functional upgrades to provide better global pandemic control, a platform for telemedicine, and decentralized testing (Fig. S12). Moreover, this versatile tool can be

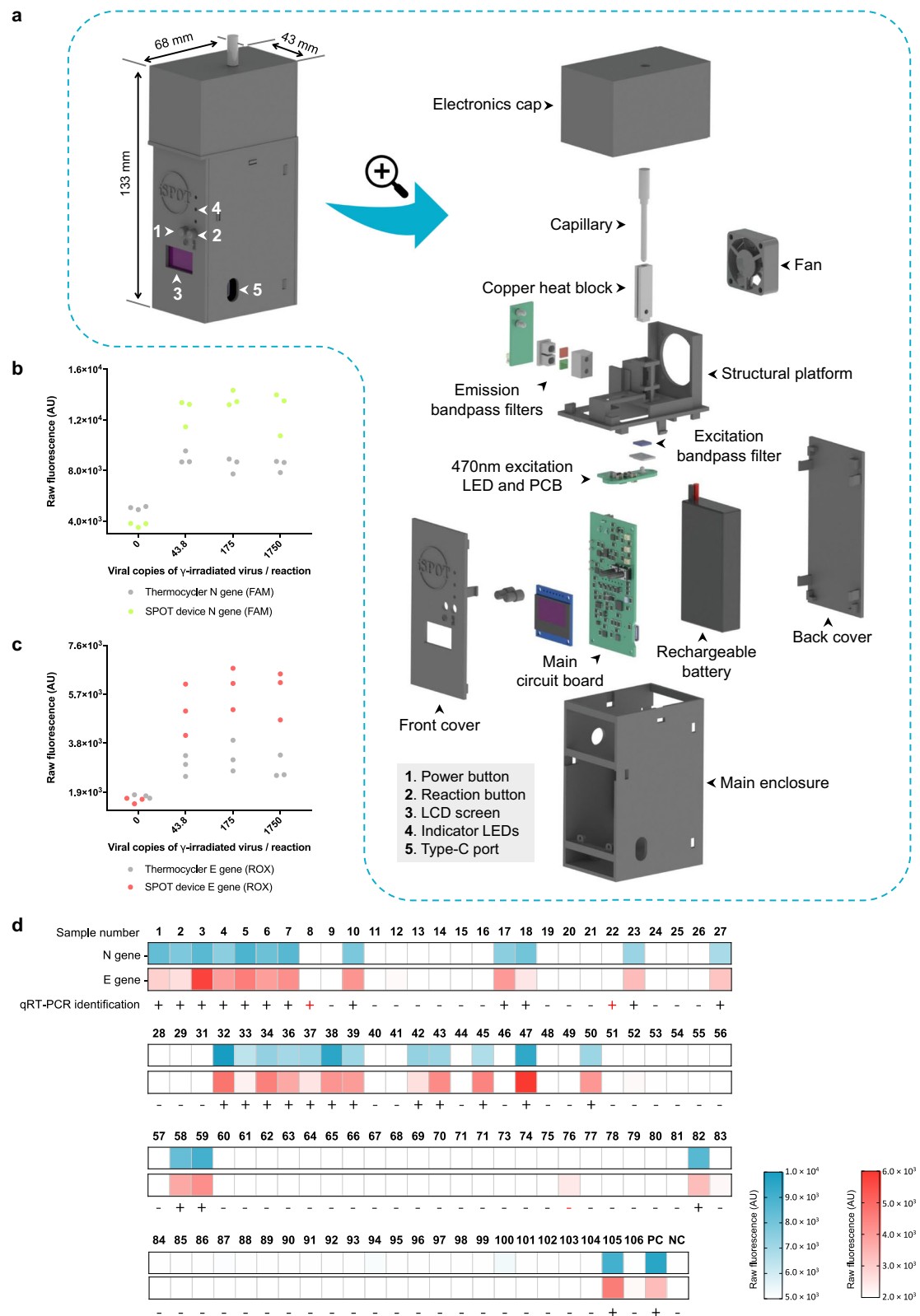

*ᵃ Ct value of qRT-PCR in Supplementary Table 1.*

developed for testing single nucleotide polymorphisms (SNPs) and other pathogens as well.

## Methods

**Specimen and nucleic acid preparation**. To evaluate the performance characteristics of the SPOT assay, three types of specimens were tested. The RNA template was synthesized by IVT. First, DNA fragments of SARS-CoV-2 (covering N gene and E gene regions) were synthesized by Integrated DNA Technologies (IDT) as IVT templates. A PCR step was performed on the synthetic gene fragment with a forward primer containing a T7 promoter. Next, the PCR product was used as the template for an IVT reaction at 37 °C for 4 h using Precision RNA Synthesis Kit (A29377, Invitrogen). The IVT reaction was then treated with DNase I for 15 min at 37 °C followed by an RNA purification step. The RNA was quantified by

**Fig. 3 Design and validation of the SPOT device. a** Rendering of the fully enclosed SPOT device along with a partially exploded view. Users press buttons to power on the device (1) and start detection reactions (2) and read results on the integrated LCD screen (3). Reaction progress is displayed on the LCD screen and indicator LEDs (4). The SPOT device can export data via a USB cable (5) and be recharged. An electronics cap covers the optical and thermal modules, which are comprised of a capillary in a copper heat block, a cooling fan, a photodiode PCB placed securely behind emission bandpass filters, a structural platform, and an excitation bandpass filter covering a 470 nm excitation LED and PCB. The reaction timings, temperature, and results interpretation are performed on a main circuit board powered by a rechargeable battery, contained within a main enclosure capped with front and back covers. **b, c** Capability of performing detection by using the SPOT device. SPOT assays using γ-irradiated virus spiked samples were run on both the SPOT device (green dots for N gene in **b** and red dots for E gene in **c**) and a thermocycler (gray dots in **b** and **c**), then fluorescence values were measured on the SPOT device. SPOT assays performed on the SPOT device using γ-irradiated virus spiked samples at 0, 43.8, 175, and 1750 viral copies per reaction ($n = 3$). **d** Validation using clinical saliva samples in a blinded test. Patient sample fluorescence data from SPOT device readout. Clinical samples from 104 patients were collected from the campus of University of Illinois at Urbana-Champaign. Signal intensities from SPOT were measured using a ×2 amplifier gain on photodiode signal, with a positive threshold set at 5× s.d. above background. "PC" indicates the positive control sample, "NC" indicates the negative control sample. Source data are provided as a Source Data file.

Nanodrop and Qubit and spiked into pooled human saliva (IRHUSL5ML, Innovative Research Inc.) to the specified concentration.

γ-irradiated SARS-CoV-2 (NR-52287, BEI Resources) virus samples were spiked into pooled human saliva and treated with QuickExtract DNA Extraction Solution (QE09050, Lucigen) at 95 °C for 5 min at a 1:1 ratio. Other respiratory viral samples including Human coronavirus OC43 (NR-52727, BEI Resources), Human coronavirus 229E (NR-52728, BEI Resources), Human Coronavirus NL63 (NR-44105, BEI Resources), MERS-Coronavirus (NR-50549, BEI Resources), SARS-CoV (NR-9324, BEI Resources) and Influenza A (NR-43023, BEI Resources) were spiked into pooled human saliva and treated with QuickExtract DNA Extraction Solution at 95 °C for 5 min at a 1:1 ratio.

Clinical saliva samples obtained from the Veterinary Diagnostic Laboratory of University of Illinois at Urbana-Champaign were treated with QuickExtract DNA Extraction Solution at 95 °C for 5 min at a 1:1 ratio. Clinical trial study was registered as an IBC project with the project number IBC-4609 and approved by the Division of Research Safety, University of Illinois at Urbana-Champaign. They have obtained informed consent from all participants.

**SPOT assay components preparation**. *Pf*Ago protein was purified using the established protocol[11]. Briefly, a strep(II)-tagged (N-terminal) codon-optimized version of *Pf*Ago gene was cloned into pET28a plasmid. The expression plasmid was transformed into *Escherichia coli* KRX (Promega) followed by induction and purification. The purified protein was stored in storage buffer (20 mM Tris-HCl, pH 8.0, 300 mM NaCl, 0.5 mM MnCl₂, 15% (v/v) glycerol) and the aliquots were stored at −80 °C.

The length and GC content optimized reporter is a key component in SPOT detection as cleavage of the reporter will generate a detectable fluorescent output signal. In order to get significant net fluorescence value increases in the detection of SARS-CoV-2 using the SPOT assay, we used a 17 nt reporter probe (N gene: /56-FAM/XXX/3BHQ-1/, E gene: /6-ROX/XXX/3BHQ-2/) for *Pf*Ago recognition and cleavage. 5' phosphorylated 16 nt gDNAs were used for primary cleavage to generate the secondary gDNA and trigger the secondary cleavage reaction. All the primers, gDNAs, and reporters were ordered from IDT, and sequences are listed in Supplementary Data 4.

**SPOT assays**. SPOT assays were performed using RT-LAMP for preamplification of IVT-RNA or viral RNA targets and *Pf*Ago for the following multiplexed cleavage assay. RT-LAMP master mix was prepared as suggested by New England Biolabs (www.neb.com/protocols/2014/10/09/typical-rt-lamp-protocol) with following changes: WarmStart® Bst 2.0 (Cat# M0538L, NEB) with the final concentration of 320 units/mL, WarmStart® RTx (Cat# M0380L, NEB) with the final concentration of 300 units/mL, and final volume of 40 μL (includes test sample). LAMP primers were added at a final concentration of 0.2 μM for F3 and B3, 1.6 μM for forward inner and backward inner primers and 0.8 μM for loop forward and loop backward primers. For duplexed LAMP reaction, the final concentration of primer was reduced to half of the concentration mentioned above for each gene, and two sets of primers (total six oligos) were used to target the SARS-CoV-2 N (nucleoprotein) and E (envelope) genes respectively. Reactions were performed using 5 μL of IVT-RNA or pretreated specimens at 63 °C for 20-30 min.

*Pf*Ago detection master mix was prepared with 1× Isothermal amplification buffer (Cat# B0537S, NEB), 1.4 μM *Pf*Ago, 0.5 mM MnCl₂, 1.25 μM of each gDNA specific to N gene and E gene (total six gDNAs), 312.5 nM FAM-labeled reporter (N gene) and 625 nM ROX-labeled reporter (E gene), and brought up to 40 μL. The concentrations indicated above are the final concentrations of each component in the total 80 μL detection reaction (40 μL RT-LAMP product plus 40 μL *Pf*Ago master mix). The RT-LAMP product was then mixed with *Pf*Ago detection master mix. As the temperature was raised to 95 °C, *Pf*Ago initiated cleavage reaction and released the fluorescent signal. For samples containing SARS-CoV-2 nucleic acids, 3–5 min incubation at 95 °C generated detectable fluorescent output.

**One-step SPOT for SARS-CoV-2 saliva sample detection**. To endow SPOT assay with the ability to be applicable at home, we established a one-step SPOT assay to simplify the liquid handling operations. Two prefabricated capillaries are needed for performing entire detection procedures. Capillary 1 was designed for saliva sample pretreatment by loading 20 μL QuickExtract DNA Extraction Solution to the bottom of the capillary and sealing with a plastic stopper. Capillary 2 was harnessed for SPOT assay by creating a two-compartment system. First, 40 μL *Pf*Ago detection reaction was loaded into the bottom of the capillary. Then, 50 mg paraffin wax powder was added into the capillary above the liquid surface followed by heating the capillary above 70 °C for 1 min to induce complete melting of the paraffin wax. After paraffin wax congealed, 35 μL LAMP reaction mixture was inserted into the capillary, on top of the paraffin wax. Paraffin wax powder can be prepared as follows: use a blade to cut some paraffin wax chips from a larger block (Cat# BW-439, Blended Waxes, Inc.), then place the chips into a mortar and add liquid nitrogen until it completely covers the chips. Grind chips gently in the mortar and collect the powder in a glass bottle.

A portable SPOT device was designed and fabricated for this one-step SPOT assay. First, a collection capillary (Cat# 1-000-0200, MICROCAPS®) was used to obtain around 20 μL of saliva from patient's mouth via automatic suction by the capillarity of tube. Then, the saliva sample was injected into capillary 1 and mixed with the lysis buffer by pressing the rubber cap. Capillary 1 was then inserted into the SPOT device followed by pressing the reaction button to initiate the lysis program (95 °C for 5 min). Following completion of the lysis program, capillary 1 was removed from the SPOT device, and a new collection capillary was used to transfer around 5–10 μL lysate into the upper compartment of capillary 2. Capillary 2 was then inserted into the SPOT device followed by pressing the reaction button to initiate the detection program (63 °C for 30 min, 95 °C for 5 min, followed by a 1-minute fan cooling period). For samples containing detectable SARS-CoV-2 nucleic acids, the test readout will display on the screen of the device with "Positive", otherwise "Negative". The result will be indicated as "Inconlusive" in samples where only one gene is detected.

**qRT-PCR analysis of clinical saliva samples**. qRT-PCR analysis of clinical saliva samples was performed by the Veterinary Diagnostic Laboratory of University of Illinois at Urbana-Champaign, using the Fisher TaqPath COVID-19 Combo kit (Applied Biosystems) and QuantStudio 3 Real-Time PCR System (Applied Biosystems). The RT-qPCR was run using the standard mode, consisting of a hold stage at 25 °C for 2 min, 53 °C for 10 min, and 95 °C for 2 min, followed by 40 cycles of a PCR stage at 95 °C for 3 s then 60 °C for 30 s; with a 1.6 °C/s ramp up and ramp down rate[16].

**Design and assembly of the SPOT device**. The SPOT device is comprised of 3D printed structural parts, a machined copper heat block, optical filters, and electrical components. CAD files of 3D printed parts and the copper heat block can be found in the supplementary materials. Parts were printed using a Form 3 (Formlabs; Somerville, MA) 3D printer with Black Resin V4 (RS-F2-GPBK-04, Formlabs). The copper heat block was machined at the University of Illinois Aerospace and Physics-MRL machine shops. PCBs were fabricated by ALLPCB (Zhejiang, China). Circuit diagrams for the SPOT electrical components and a list of all electrical parts used for PCB assembly are available in the supplementary information. Optical filters were purchased from Salvo Technologies (Pinellas Park, Florida). The excitation LED was filtered by a 470 nm maximum with 20 nm full width at half maximum (FWHM) bandpass filter (SKU: CO674-91, Salvo Technologies). For quantification of ROX fluorescence, a 610 nm maximum 20 nm FWHM bandpass filter was used (SKU: CO674-65, Salvo Technologies). For quantification of FAM fluorescence, a 520 nm maximum 20 nm FWHM bandpass filter was used (SKU: 102387140, Salvo Technologies).

The motherboard and battery are mounted into the bottom enclosure, which has openings for the charging cable, USB cable, power and start buttons, an LCD screen, and indicator LEDs. The main platform fits on the top of the bottom enclosure and

houses the thermal and optical modules. The optical module consists of a blue LED (710-150353BS74500, Mouser; Mansfield; TX) beneath the excitation bandpass filter which shines into a hole at the bottom of the copper heat block, directly into a capillary. Two emission bandpass filters are mounted perpendicular to the capillary and filter light passing to the photodiodes. A transimpedance amplifier is then used to convert photodiode current into voltage to be read by the microcontroller. The user-selected gain is then used to configure the programmable-gain amplifier which results in increased dynamic range. The copper heat block is wrapped with fiberglass insulated nichrome wire and heats upon application of an electrical current. After completion of the reaction, an integrated fan turns on for rapid cooling. To block excess light, a cap covers the platform section containing a circular opening for insertion and removal of capillaries. The CAD file for the entire SPOT device assembly can be found in the supplementary materials.

Using the SPOT dashboard software, a computer can be used to monitor and configure the SPOT device, quantify sample fluorescence, and export data. With the configuration utility, the time and temperature for the LAMP and *Pf*Ago cleavage reactions can be modified if desired, which by default are set to 63 °C for 30 min and 95 °C for 5 min, respectively. User guides for the SPOT device and dashboard software can be found in the supplementary information.

**Statistical analysis**. Data is shown as original data or the mean with error bars representing standard deviations. Graphs were made by using the GraphPad Prism software (version 9.0.0).

**Reporting summary**. Further information on research design is available in the Nature Research Reporting Summary linked to this article.

## Data availability
All data is available in the main text or the supplementary information text and files. Fluorescent signal data from the SPOT device and qPCR machine are available from the corresponding author upon request, and can be found at Figshare (https://doi.org/10.6084/m9.figshare.14417639.v1) as well. Source data are provided with this paper.

Reagents and consumables described in this work are available from commercial sources. *Pf*Ago was purified following the established protocol[11], as summarized above. A step-by-step experimental protocol is shared at Protocols.io (https://doi.org/10.17504/protocols.io.bujvnun6). Any other relevant data are available from the authors upon reasonable request. Source data are provided with this paper.

## Code availability
The custom codes used in this study, including but not limited to the SPOT device microcontroller, display, fluorescent quantification, and data analysis, are available from the corresponding author upon request. Codes can also be accessed on GitLab following this link (https://gitlab.engr.illinois.edu/ispot). A permanent reference to the version of the SPOT code at the time of publication can be found at https://doi.org/10.5281/zenodo.4695139.

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

## Acknowledgements
This work was supported by Steve L. Miller Endowed Chair Fund and Carl R. Woese Institute for Genomic Biology at University of Illinois at Urbana-Champaign. The clinical saliva samples were obtained from University of Illinois at Urbana-Champaign COVID-19 Registry and we thank Dr. Leyi Wang for preparing these samples. We thank the University of Illinois at Urbana-Champaign Aerospace and Physics-MRL machine shops for assistance in creating the heat block used in the SPOT device.

## Author contributions
G.X., S.L., and H.Z. conceived and designed the study; G.X. conceived, designed, and validated SPOT reagents and protocols. S.L., V.P., and B.P. designed and assembled the device. G.X. and S.L. performed experiments and analyzed data. G.X. and S.L. validated SPOT assay on clinical saliva samples. G.X., S.L., and H.Z. wrote and edited the manuscript. H.Z. supervised and discussed with all authors during this study. All authors read the manuscript and agreed to its contents.

## Competing interests
All authors declare the following competing interests. This study has been submitted to the United States Patent and Trademark Office (USPTO) as an invention patent application by H.Z., G.X., and S.L. based on the results of this study (U.S. Patent Application No. 63/155,127). Status of application: Provisional Specific aspect of manuscript covered in patent application: Combination of RT-LAMP with *Pf*Ago detection of RNA viruses; Development of a portable handheld diagnostic device.
