## [Peer Review File · Nature Communications]

Reviewers' Comments:

Reviewer #1:

Remarks to the Author:

In this manuscript by Xun, et al., the authors present a rapid Scalable and Portable Testing (SPOT) system consisting of an RT-LAMP assay, PfAgo-based target detection, and a battery-powered portable device. They were able to detect as low as 7.5 copies per reaction of IVT SARS-CoV-2 RNA. They analyzed 104 clinical saliva samples and demonstrated 28/30 (93.% sensitivity) and 73/74 (98.6% specificity).

The advantages of the SPOT system include – quick extraction bypassing a time-consuming nucleic acid extraction step, one-pot reaction, speed (<1 hour?), use of the PfAgo-based target detection which allows multiplexing, and a battery-powered device. The performance appears to be excellent, and comparable with PCR-based assays. However, there are some issues with this manuscript, as follows:

- There needs to be more rigorous evaluation of the performance characteristics of the test. The authors should use the FDA guidance for COVID-19 molecular testing on this topic: <https://www.fda.gov/medical-devices/emergency-situations-medical-devices/coronavirus-covid-19-and-medical-devices>. In particular, a rigorous limit of detection should be performed using 95% probit analysis, a specificity panel versus other coronaviruses and influenza viruses should be run, etc.
- The overall workflow and turnaround time for the assay remains unclear to me. How fast can the test be run? There is a 5-minute heating step and at least 20-minute RT-LAMP step right?
- The use of two capillary transfer steps suggests that this would likely be useful in POC settings such as emergency rooms and clinics but not at home, since it probably takes a certain level of expertise to run the assay.
- * Can the authors comment on how far the costs for the device can go down with scaling volumes? \$222 seems expensive to me. Also, are their considerations with regards to limited reagents?
- Abstract – instead of 93.3% accuracy and 98.6% accuracy, the more precise term is 93.3% sensitivity and 98.6% specificity

Reviewer #2:

Remarks to the Author:

Xun et al. describe the development of a nucleic acid detection tool: a Scalable and Portable Testing (SPOT) system for COVID-19 diagnosis. The key component of the diagnosis approach is a DNA-guided Argonaute nuclease of the hyperthermophilic archaeon *Pyrococcus furiosus* (PfAgo). A (RT)-LAMP pre-amplification step is coupled to a PfAgo-based target sequence detection reaction, resulting in a detectable fluorescence output. Addition of the developed PfAgo-based sequence detection increases the specificity, allowing for differentiation of a true positive and a false positive, that could be the result of non-specific amplification when (RT)-LAMP is used by itself. For the development of the SPOT assay, SARS-CoV-2 was chosen as a proof of principle target. The SPOT assay was optimized by choosing a set of multiple gDNAs to improve efficiency in the initial "guide releasing" cleavage, resulting in a higher fluorescent signal. The limit of detection on synthetic target material was determined for two different target genes, which were finally successfully combined into a single, multiplexed reaction. Because it was found that the (RT)-LAMP pre-amplification step was inhibited by the PfAgo reaction components, a capillary encapsulation method was developed, having 2 separate chambers with paraffin wax in between. A prototype device was made to perform the sample extraction step and subsequent single-tube multistep

detection reaction. Eventually, the SPOT system was successfully used for the detection of SARS-CoV-2 in clinical saliva samples. The manuscript is generally well written and well presented (but, see comments).

Comments

1. Abstract - readability of Abstract will be improved to reduce the amount of details (e.g. numbers).
2. Results - It is implied in the results section that a quantitative test result is given, this however is not shown in a convincing manner. Please improve.
3. Figure 1b should be self-explanatory, but it is not. Please replace by panel of Suppl Fig S1a, this would solve the problem.
4. Page 3, lines 1-6 - Results of fluorescent read-out are shown on LCD screen - what about the background, is that subtracted from the presented value, or not ? Please add this detail in the text.
5. Page 3, line 12-18 - The description of the results does not match well with the presented experimental data, in other words, I suggest that the interpretations of the performance of the diagnostics approach should be toned down (throughout manuscript). One example: "We compared the cleavage efficiency in target gene detection reactions containing two or three primary gDNAs and found use of three gDNAs "dramatically enhanced" the efficiency of cleavage (...) (Figure S1c)." The scale of Fig S1c should be adjusted, to show clearly that the improvement is only 2-3 fold. Please rephrase to "dramatically enhanced" > "2-3 fold enhanced".
6. Page 3, line 12. "we optimized the GC content and length of the reporter to minimize the unwanted background fluorescence (data not shown)." - would be good to show these relevant data as Supplementary Data.
7. Page 3, Figure S1c legend - "Comparison of cleavage efficiency between combinations of gDNAs (n = 3). (d) Optimization of the fluorescence-based nucleic acid detection reaction. Three candidate enzymes were added at the same amount. TtAgo mediated reaction was incubated at 80°C without manganese ion (Mn²⁺). PfAgo mediated reaction was performed at 95°C with manganese ion (Mn²⁺). Optimized PfAgo mutant, which shows cleavage activity in the presence of magnesium ion (Mg²⁺), was added to the reaction at 95°C (n = 3)." Again, however, the experimental set-up is in favour of PfAgo, not of TtAgo; what is the rationale for performing the assay under these conditions, given that substantially enhanced activity is to be expected at 70°C in the presence manganese (Swarts 2014). The authors should repeat the experiment with TtAgo under optimal conditions, or provide a motivation on their choice of these sub-optimal conditions.
8. General - Argonaute-based approach is compared to CRISPR-Cas -based approaches. Specificity is an important criterion. It would be important to reveal this important feature of Argonaute in general, and of PfAgo in particular. Please compare with Cas nucleases. If data is not available from previously published studies, a single/double mismatch screen along the guide should be performed.

Minor comments

9. Page 3, line 33 - "... with the LoD." Please rephrase this (part of the) sentence.
10. Page3, line 34 - "To minimize false positives and achieve high accuracy ..."
11. Page 3- line 41 - both two - Typo

NCOMMS-20-51409

Title: A Rapid, Accurate, Scalable and Portable Testing (SPOT) System for COVID-19 Diagnosis

Response to the editor regarding reviewer comments

We wish to express our sincere gratitude to the editor and reviewers for their time and consideration of this manuscript. We have revised the manuscript substantially and included additional other respiratory viruses to systematically evaluate the performance characteristics of our strategy to address the reviewers' concerns as well as improve the overall content of the paper. In this document, we address the reviewers' concerns, which are in black text. Our comments to the reviewers are in blue text. Sections that have been copied and pasted from the new manuscript, which have been added or edited to specifically address the reviewers' comments, are in orange text.

Overall summary of changes:

We thank the editor and the two reviewers for carefully reading our manuscript and providing valuable suggestions. We believe that these suggestions significantly improved the overall quality of our manuscript. Below, we provide our point-by-point responses and highlight additional data that has been included in the manuscript. Following Reviewer #1's recommendation to increase confidence in our data, we added a specificity panel of 6 additional respiratory viruses, including coronaviruses and the influenza virus. The results show a high specificity of the SPOT assay, consistent with the results described in the previous manuscript. We also included the GC content optimization data in the supplementary information to support our conclusion in the main text.

Response to Reviewer #1

Reviewer #1 (Remarks to the Author):

In this manuscript by Xun, et al., the authors present a rapid Scalable and Portable Testing (SPOT) system consisting of an RT-LAMP assay, PfAgo-based target detection, and a battery-powered portable device. They were able to detect as low as 7.5 copies per reaction of IVT SARS-CoV-2 RNA. They analyzed 104 clinical saliva samples and demonstrated 28/30 (93.% sensitivity) and 73/74 (98.6% specificity).

The advantages of the SPOT system include – quick extraction bypassing a time-consuming nucleic acid extraction step, one-pot reaction, speed (<1 hour?), use of the PfAgo-based target detection which allows multiplexing, and a battery-powered device. The performance appears to be excellent, and comparable with PCR-based assays. However, there are some issues with this manuscript, as follows:

- There needs to be more rigorous evaluation of the performance characteristics of the test. The authors should use the FDA guidance for COVID-19 molecular testing on this topic: <https://www.fda.gov/medical-devices/emergency-situations-medical-devices/coronavirus-covid-19-and-medical-devices>. In particular, a rigorous limit of detection should be performed using 95% probit analysis, a specificity panel versus other coronaviruses and influenza viruses should be run, etc.

We thank the reviewer for the suggestion to systematically evaluate the performance characteristics by following the FDA guidance. In our revised manuscript, we included 6

respiratory viruses, including Human coronavirus OC43, Human coronavirus 229E, Human Coronavirus NL63, MERS-Coronavirus, SARS-CoV and Influenza A, to test the specificity of our assay. By using the protocol described in the main text, we could only obtain a positive signal from our positive control sample containing SARS-CoV-2 nucleic acids. This result demonstrates that the specificity of our system is high enough to distinguish SARS-CoV-2 from other respiratory viruses. Also, we established a regression model in an attempt to perform 95% probit analysis, however we were unable to obtain datapoints between undetectable viral concentrations and our reported LoD. The figure provided below is a regression curve based on our data. As an alternative method, we ran another 30 replicates with our reported LoD concentration, the results revealed that our reported LoDs were reliable and reproducible. The data was added into the supplementary information as Figure S7. Note that this same method was used in several other CRISPR based SARS-CoV-2 detection papers. We cited a paper from Feng Zhang's group here (Point-of-care testing for COVID-19 using SHERLOCK diagnostics, <https://doi.org/10.1101/2020.05.04.20091231>) for your reference.

Page 3 lines 43-45; page 4 lines 1-3:

In order to validate the specificity of our detection system to SARS-CoV-2, saliva was spiked with or without SARS-CoV-2, three other human coronaviruses (OC43, 229E and NL63) genomic RNA, SARS and MERS viruses (γ -irradiated), and influenza A genomic RNA. Among these samples, SARS-CoV-2 genes were only detected in the positive control, further supporting the specificity of the detection platform for SARS-CoV-2 (Figure S8).

Supplementary Figure 8. Specificity of SPOT assay on SARS-CoV-2 among other respiratory viruses. Commercially available saliva samples (Innovative Research) were spiked with human coronaviruses (OC43, 229E and NL63) genomic RNA, SARS and MERS viruses (γ -irradiated), and influenza A genomic RNA, and diluted 1:1 with QuickExtract DNA Extraction Solution. Samples were heat treated at 95°C for 5 min. Virus-spiked saliva samples, a positive control (γ -irradiated SARS-CoV-2) and a negative control were incubated on a thermocycler and measured using a qPCR machine (QuantStudio 3 RT-PCR system) (n = 3).

Supplementary Figure 7. LoD reliability and reproducibility of SPOT assay. (a) 30 replicates of *in vitro* transcribed RNA spiked saliva samples with the LoD concentration (7.48 copies/reaction) were detected by SPOT assay. (b) 30 replicates of γ -irradiated virus spiked saliva sample with the N gene LoD concentration (17.5 copies/reaction) were detected by SPOT assay. (c) 30 replicates of γ -irradiated virus spiked saliva sample with the E gene LoD concentration (43.8 copies/reaction) were detected by SPOT assay. All samples were incubated on thermocycler and measured on a qPCR machine (QuantStudio 3 RT-PCR system). Error bar represents the mean with 95% CI.

- The overall workflow and turnaround time for the assay remains unclear to me. How fast can the test be run? There is a 5-minute heating step and at least 20-minute RT-LAMP step right?

We thank the reviewer for raising this point. In our detection assay, the overall workflow is in three steps: (1) 5-minute heating at 95°C for RNA quick extraction; (2) at least 20-minute heating at 63°C for RT-LAMP; (3) 3-minute heating at 95°C for *PfAgo*-mediated detection. Thus, the total turnaround time for this assay is 28 minutes for a sample with an original viral number above 50 copies. For the sample which has original viral number below 50 copies, especially relevant to clinical testing, we extended the RT-LAMP time to 30 minutes and the *PfAgo* detection time to 5 minutes to ensure consistent detection of positive samples.

- The use of two capillary transfer steps suggests that this would likely be useful in POC settings such as emergency rooms and clinics but not at home, since it probably takes a certain level of expertise to run the assay.

We thank the reviewer for the thoughtful suggestion on the application scenarios of our detection system. Since the transfer step does not have to be particularly precise, and the capillary absorbs a certain amount of sample liquid due to its capillarity, we are confident the assay can be performed with only minimal training. In fact, we recruited some students from our laboratory to test the feasibility that the SPOT system can be used for at-home detection by following the instructions of our user manual. The recruited students were able to accurately perform the SPOT testing system per our step-by-step protocol without

professional supervision. Thus, we believe that the SPOT system will be suitable for at-home testing.

* Can the authors comment on how far the costs for the device can go down with scaling volumes? \$222 seems expensive to me. Also, are their considerations with regards to limited reagents?

We agree that \$222 is a steep price for a single device. Considering that the majority of the SPOT device's price tag is comprised of optical filters, we reached out to our chosen vendor to inquire about the price reduction with a large-scale purchase of filters. The \$222 price tag originally mentioned in our manuscript included \$153 of optical filters per device, assuming purchase of only 10 filters. The cost of filters drops to only \$9.05 per device at large-scale purchase of 10,000 filters, bringing the cost of each device to under \$78. As our bill of materials assumes purchasing 100 of each electronic component, large-scale production of 10,000 devices would also warrant negotiation on prices of each component with the original manufacturers, opening the door for a further substantial drop in price.

Page 5 lines 28-29:

The material costs would sum to less than \$78 per device at scale production of 10,000 devices (Supplementary File 2).

- Abstract – instead of 93.3% accuracy and 98.6% accuracy, the more precise term is 93.3% sensitivity and 98.6% specificity.

We thank the reviewer for pointing out these errors, which have been revised in the updated manuscript.

Response to Reviewer #2

Reviewer #2 (Remarks to the Author):

Xun et al. describe the development of a nucleic acid detection tool: a Scalable and Portable Testing (SPOT) system for COVID-19 diagnosis. The key component of the diagnosis approach is a DNA-guided Argonaute nuclease of the hyperthermophilic archaeon *Pyrococcus furiosus* (PfAgo). A (RT)-LAMP pre-amplification step is coupled to a PfAgo-based target sequence detection reaction, resulting in a detectable fluorescence output. Addition of the developed PfAgo-based sequence detection increases the specificity, allowing for differentiation of a true positive and a false positive, that could be the result of non-specific amplification when (RT)-LAMP is used by itself. For the development of the SPOT assay, SARS-CoV-2 was chosen as a proof of principle target. The SPOT assay was optimized by choosing a set of multiple gDNAs to improve efficiency in the initial “guide releasing” cleavage, resulting in a higher fluorescent signal. The limit of detection on synthetic target material was determined for two different target genes, which were finally successfully combined into a single, multiplexed reaction. Because it was found that the (RT)-LAMP pre-amplification step was inhibited by the PfAgo reaction components, a capillary encapsulation method was developed, having 2 separate chambers with paraffin wax in between. A prototype device was made to perform the sample extraction step and subsequent single-tube multistep detection reaction. Eventually, the SPOT system was

successfully used for the detection of SARS-CoV-2 in clinical saliva samples. The manuscript is generally well written and well presented (but, see comments).

Comments

1. Abstract - readability of Abstract will be improved to reduce the amount of details (e.g. numbers).

We thank the reviewer for the valuable suggestion. We have revised the abstract to improve its readability by removing some non-critical numbers.

2. Results - It is implied in the results section that a quantitative test result is given, this however is not shown in a convincing manner. Please improve.

We thank the reviewer for pointing out this issue. Although we have observed quantitative results in a small dynamic range (in the range of approximately 20-40 viral copies per reaction), we agree that this is not well displayed by the results presented in this manuscript. Moreover, even if we were to add additional data demonstrating quantitative test results, the small dynamic range does not adequately cover the range of viral copies expected within saliva samples from SARS-CoV-2 positive patients. Although the SPOT device is capable of fluorescence quantification, we have revised our manuscript to remove any implication of a quantitative assay result.

Page 2 line 2: Removed the following sentence:

Moreover, as test results are read from a lateral flow strip, these systems produce a qualitative rather than quantitative result.

Page 3 line 4: "quantitative" was deleted from "and quantitative test results are displayed on an attached LCD screen"

Page 11 line 28: "quantifiable" changed to "detectable"

The reporter probe is a key component in SPOT detection as cleavage of the probe will generate a detectable fluorescent output signal.

Page 13 line 2: "quantifiable" changed to "detectable"

For samples containing SARS-CoV-2 nucleic acids, 3-5 min incubation at 95°C generated detectable fluorescent output.

3. Figure 1b should be self-explanatory, but it is not. Please replace by panel of Suppl Fig S1a, this would solve the problem.

We thank the reviewer for the valuable suggestion. As the mechanism of *PfAgo*-based nucleic acid detection was reported by Xun et al. previously (doi: <https://doi.org/10.1101/821280>), we alternatively added a simplified schematic of principle of *PfAgo* mediated detection in the main text. To better clarify, we cited Figure S1 in the relevant section of the main text (page 3, line 11) to ensure understanding by readers who are not familiar with the mechanism of *PfAgo*-based nucleic acid detection.

Figure 1b:

PfAgo cleavage sites are shown in black triangles, enclosed “Q” represents quencher group and enclosed “F” represents fluorophore group.

4. Page 3, lines 1-6 - Results of fluorescent read-out are shown on LCD screen - what about the background, is that subtracted from the presented value, or not? Please add this detail in the text.

We thank the reviewer for raising this point. We calibrated our SPOT device with the negative control samples as the background signal and set the positive threshold at $5 \times$ s.d. above background in the device's program.

Page 3 lines 5-6:

The SPOT device is calibrated using negative control samples to quantify the background signal with the positive threshold set to $5 \times$ s.d. above background in the device's program.

5. Page 3, line 12-18 - The description of the results does not match well with the presented experimental data, in other words, I suggest that the interpretations of the performance of the diagnostics approach should be toned down (throughout manuscript). One example: "We compared the cleavage efficiency in target gene detection reactions containing two or three primary gDNAs and found use of three gDNAs "dramatically enhanced" the efficiency of cleavage (...) (Figure S1c)." The scale of Fig S1c should be adjusted, to show clearly that the improvement is only 2-3 fold. Please rephrase to "dramatically enhanced" > "2-3 fold enhanced".

We thank the reviewer for this suggestion. We have revised the manuscript as per your suggestion.

Page 3 lines 16-18:

We compared the cleavage efficiency in target gene detection reactions containing two or three primary gDNAs and found use of three gDNAs enhanced the efficiency of cleavage, which led to a 2-3 fold higher fluorescent signal than use of only two gDNAs (Figure S2b).

6. Page 3, line 12. "we optimized the GC content and length of the reporter to minimize the unwanted background fluorescence (data not shown)." - would be good to show these relevant data as Supplementary Data.

We thank the reviewer for the valuable suggestion. We have added this part of data in the supplementary information (Figure S2a) to support the conclusion mentioned in our main text.

Figure S2a:

(a) Optimization of GC content of the fluorescent reporters. The GC contents of three fluorescent reporters are 76%, 59% and 47%, respectively. Fluorescent signals were collected after 1 min, 3 min, and 5 min *PfAgo* cleavage with the same amount of LAMP products input ($n = 3$). The net fluorescence value increases were calculated by subtracting the initial background fluorescence value in order to reduce the difference in fluorescence intensity between different reporters.

7. Page 3, Figure S1c legend - "Comparison of cleavage efficiency between combinations of gDNAs ($n = 3$). (d) Optimization of the fluorescence-based nucleic acid detection reaction. Three candidate enzymes were added at the same amount. TtAgo mediated reaction was incubated at 80°C without manganese ion (Mn^{2+}). *PfAgo* mediated reaction was performed at 95°C with manganese ion (Mn^{2+}). Optimized *PfAgo* mutant, which shows cleavage activity in the presence of magnesium ion (Mg^{2+}), was added to the reaction at 95°C ($n = 3$)." Again, however, the experimental set-up is in favour of *PfAgo*, not of *TtAgo*; what is the rationale for performing the assay under these conditions, given that substantially enhanced activity is to be expected at 70°C in the presence manganese (Swarts 2014). The authors should repeat the experiment with *TtAgo* under optimal conditions, or provide a motivation on their choice of these sub-optimal conditions.

We thank the reviewer for pointing this concern out. To clarify, Swarts 2014 employed their cleavage assay at 75°C for both ssDNA and dsDNA cleavage. However, our assay employed commercial *TtAgo* protein (#M0665S) from New England Biolabs Inc. (NEB), which recommends cleavage to be performed at 80°C according to the NEB protocol. In addition, we believe higher temperatures promote release of secondary gDNAs and improve the cleavage cascade.

8. General - Argonaute-based approach is compared to CRISPR-Cas -based approaches. Specificity is an important criterion. It would be important to reveal this important feature of Argonaute in general, and of *PfAgo* in particular. Please compare with Cas nucleases. If data is not available from previously published studies, a single/double mismatch screen along the guide should be performed.

We thank the reviewer for pointing out this critical feature. However, the specificity of *PfAgo* cleavage has been thoroughly investigated in a previous report, “A-Star, an Argonaute-directed System for Rare SNV Enrichment and Detection” (doi: <https://doi.org/10.1101/803841>), which systematically characterized the specificity of *PfAgo* cleavage. The results revealed that *PfAgo* can recognize and discriminate single mismatch to selectively cleave the target sequence by introducing additional mismatched nucleotide in the cleavage position (gDNA nucleotide position 10 or 11) of gDNA. Therefore, we believe that *PfAgo* has its unique property to perform the detection accurately. Moreover, due to the nonspecific cleavage of target ssRNA/ssDNA reporter by the collateral effect of Cas 12a, Cas12b or Cas13a, Cas-based detection assays cannot be developed as single enzyme mediated multiplexed detection methods.

Minor comments

We thank the reviewer for these valuable suggestions. We have revised those sentences in our main text.

9. Page 3, line 33 - “... with the LoD.” Please rephrase this (part of the) sentence.

Fixed. Page 3 lines 31-33:

Based on the time-course assay, we found that 5 min *PfAgo* cleavage is sufficient to obtain the obvious fluorescent signal increases for detecting contrived saliva samples with the LoD viral load (Figure S4).

10. Page3, line 34 - “To minimize false positives and achieve high accuracy ...”

Fixed. Page 3 lines 34-36:

To achieve high accuracy of our assay, we next sought to detect two viral genes simultaneously in the same reaction targeting N gene region (US CDC assay) and E gene region (World Health Organization (WHO) assay)^{1,20}.

11. Page 3- line 41 - both two - Typo

Fixed. Page 3 line 43:

...that both viral genes can be amplified and detected successfully (Figure 2d).

Reviewers' Comments:

Reviewer #1:

Remarks to the Author:

In this improved manuscript by Xun, et al., the authors have addressed all of my questions raised in the initial review. I have no further comments and believe that the SPOT system will be a valuable contribution to the literature.

[

Reviewer #2:

Remarks to the Author:

Comments raised by me have been addressed appropriately.

NCOMMS-20-51409A

Title: A Rapid, Accurate, Scalable and Portable Testing (SPOT) System for COVID-19 Diagnosis

Response to Reviewer #1

Reviewer #1 (Remarks to the Author):

In this improved manuscript by Xun, et al., the authors have addressed all of my questions raised in the initial review. I have no further comments and believe that the SPOT system will be a valuable contribution to the literature.

We thank the reviewer for carefully reading our revised manuscript and providing valuable comments.

Response to Reviewer #2

Reviewer #2 (Remarks to the Author):

Comments raised by me have been addressed appropriately.

We wish to express our sincere gratitude to the reviewer for his/her time and constructive comments to improve the quality of our manuscript.